# Mapping evidence on barriers to and facilitators of diagnosing noncommunicable diseases (NCDs) among people living with human immunodeficiency virus (PLWH) in low- and middle-income countries (LMICs) in Africa: A scoping review protocol

**Abebe Sorsa Badacho** [1,2,3]*, **Ozayr Harron Mahomed** [2,4]

1 School Public Health, Wolaita Sodo University, Soddo, Ethiopia, 2 School of Nursing and Public Health, Public Health Medicine Discipline, The University of KwaZulu-Natal, Durban, South Africa, 3 Health Economics and HIV and AIDS Research Division (HEARD), University of KwaZulu-Natal, Durban, South Africa, 4 Dasman Diabetes Research Institute, Kuwait City, Kuwait

* Abebe.Badacho@wsu.edu.et, 219098944@ukzn.ac.za, sorsabebe@gmail.com

## Abstract

### Background

Noncommunicable diseases (NCDs) represent a global public health challenge in all population groups, but the prevalence of major NCDs, such as depression, hypercholesterolemia, hypertension, obesity and diabetes, is increasing at a rapid rate among people living with human immunodeficiency virus (PLWH). Studies show that integrated NCDs and human immunodeficiency virus (HIV) services have improved the patient outcome of PLWH with comorbidities with NCDs. It requires a strengthened and sustainable way of diagnosing major NCDs early among PLWH. However, there is limited evidence regarding the barriers to and facilitators of early diagnosis of NCDs (depression, hypercholesterolemia, hypertension, obesity and diabetes) among PLWH in low- and middle-income countries (LMICs). This review will systematically map the literature and describe the barriers and facilitators of diagnosing NCDs (depression, hypercholesterolemia, hypertension, obesity and diabetes) among PLWH in LMICs in Africa.

### Methods

The methodology presented by Arksey and O'Malley and the recommendation set out by Levac and colleagues will be used. PubMed, MEDLINE with full text via the EBSCO host, Google Scholar, Science Direct and Scopus will be used for a comprehensive search for data extraction. The search will be conducted using keywords, Boolean terms, and Medical Subject Headings (MeSH). All studies with eligible titles will be exported to the EndNote citation manager for deduplication and imported to Rayyan software for screening. Two reviewers will independently screen abstracts, and the preferred reporting items for systematic

**Data Availability Statement:** No datasets were generated or analyzed during the current study. All relevant data from this study will be made available upon study completion.

**Funding:** The author(s) received no specific funding for this work.

**Competing interests:** The authors have declared that no competing interests exist.

**Abbreviations:** MMAT, Mixed methods appraisal tools; **PRISMA**, Preferred Reporting Items for Systematic Review and Meta-Analysis; **PRISMA-SCR**, Preferred Reporting Items for Systematic Review and Meta-Analysis extension for scoping reviews.

review and meta-analysis extension to scoping review (PRISMA-Sc) guidelines will be used to guide the search in conducting the scoping review. We will perform descriptive data analysis for quantitative studies, NVivo software version 12 will be used for qualitative studies, and thematic content analysis will be carried out. This scoping review will include studies that included PLWH with those diagnosed with major NCDs (depression, hypercholesterolemia, hypertension, obesity, and diabetes) in LMICs in Africa.

## Discussion

This scoping review will highlight evidence mapping on barriers and facilitators of diagnosing NCDs (depression, hypercholesterolemia, hypertension, obesity, and diabetes) among PLWH LMICs in Africa.

Scoping Review Registration: registered on Open Science Framework (https://osf.io/xvtwd/).

## Introduction

Globally approximately 40 million people are living with human immunodeficiency virus (PLWH [1, 2]. The number of PLWHs has increased by more than 10 million since 2010 [2]. Sub-Saharan African countries are home to more than two-thirds (67%) of PLWHs [1, 2].

More than 28 million people currently access antiretroviral therapy (ART), more than 20 million increments from 7.8 access in 2010 [2]. Increased access to ART services has halted an estimated 12.1 million acquired immunodeficiency syndrome (AIDS) human immunodeficiency virus (HIV) related deaths since 2010 [3]. PLWHs across the globe now live longer due to improvements in and expanded access to ART [4–7].

Noncommunicable diseases (NCDs) represent a global public health challenge in all populations, leading to the deaths of more than 41 million people each year (accounting for 71% of deaths globally) [8]. Along with increased life expectancy, NCD susceptibility comes with an increased burden of NCDs [9]. Moreover, exposure to ART increases the risk of hyperlipidemia and diabetes; HIV inflammatory effects, ART drug side effects, and the risk associated with increasing age can be a risk for NCD for PLWH [10]. Studies have shown a higher incidence of NCDs among PLWH than HIV-negative patients [11]. Some HIV medicines can increase blood glucose levels and diabetes [10].

A systematic review and meta-analysis of studies conducted in low- and middle-income countries (LMICs) showed pooled estimates for the prevalence of NCDs among PLWH; the prevalence of depression, hypercholesterolemia, hypertension, andobesity was 24.4%, 22.2%, 21.2%, and 7.8%, respectively, and diabetes ranged from 1.3–18% [12]. There is a substantial burden of comorbid NCDs among PLWH in LMICs [13]. Quality of health and life gains made against HIV/AIDS are now under threat as NCDs are significantly increasing among PLWH in LMICs [14, 15]; the comorbidity of HIV and NCDs threatens to reduce global mortality due to HIV [16]. A growing number of PLWH over 50 are at higher risk for developing chronic NCDs [17].

Improved life expectancy, increased vulnerability and epidemiological transition of diseases brought a substantial burden of NCDs among PLWH in LMICs. Consequently, PLWH may now need regular screening for early detection and management of NCDs [18, 19].

However, LMICs struggle to manage growing chronic NCDs. There have been calls for the integration of HIV and NCD services to increase efficiency and improve coverage of NCD

care, although evidence of effectiveness remains unclear [20]. The increasing burden of comorbidity of HIV infection and NCDs requires a focus on healthcare services providing care for chronic multimorbidities in an integrated manner [21]. Furthermore, the demands of PLWHs with multimorbidities are significant since they are more prone to care fragmentation due to the participation of several health providers [22]. Thus, the multimorbidity of non-HIV NCDs in PLWHs has highlighted the importance of improving NCD care in this population [23]. With the growing burden of NCDs and infectious diseases in both the general population and among PLWH, healthcare systems must be strengthened for early detection and management of NCDs and making services accessible.

Strengthening and sustaining NCD services in resource-limited settings in primary healthcare is a need focus. However, there is a lack of evidence on the barriers to and facilitators of diagnosing major NCDs among PLWH. Therefore, the current study will systematically map the literature and describe the evidence on the barriers to and facilitators of the diagnosis of major NCDs, focusing on depression, hypercholesterolemia, hypertension, obesity and diabetes for PLWH in LMICs.

## Methods

This proposed scoping review will be conducted based on the framework outlined by Arksey and O'Malley [24]. It will follow five steps: step 1: Identifying the research question, step 2: Searching relevant studies, step 3: selecting appropriate studies, step 4: Charting the data, and step 5: collating, summarizing, and reporting results. The optional step 6, consultation with relevant stakeholders, has been excluded from this review. Incorporating recommendations made by Levac and colleagues [25] will be considered. A scoping study approach enables systematically searching for literature, selecting and examining literature, knowledge synthesis and mapping concepts, and the range of evidence to address an exploratory research question [24].

The Preferred Reporting Items for Systematic Reviews extension for Scoping Reviews (PRISMA-ScR) proposed by Tricco *et al.* [26] PRISMA-SCR provides a reporting guideline containing 20 essential items and two optional items that should be included in scoping reviews [26]. This guideline also facilitates methodological transparency and acceptance of research findings [26]. This protocol has been registered prospectively on the open science Frameworks (https://osf.io/xvtwd/)

### Step 1: Identifying the research question

This scoping review seeks to answer the following questions: what evidence exists on barriers and facilitators of diagnosing major NCDs focusing on depression, hypercholesterolemia, hypertension, obesity and diabetes in LMICs? We used the people concept and context (PCC) framework approach to identify the question (Table 1).

The sub questions will include the following:

**Table 1. PCC (people, concept and context) used to identify review questions.**

| | |
|---------|---|
| People | Studies include People living with HIV |
| Concept | Barriers to the diagnosis of NCDs focusing on depression hypercholesterolemia, hypertension, obesity and diabetes, Routine Screening, early detection of depression, hypercholesterolemia, hypertension, obesity and diabetes among PLWH Facilitators of diagnosing NCDs focusing on depression, hypercholesterolemia, hypertension, obesity and diabetes in LMICs among PLWH |
| Context | Low- and middle-income countries in Africa (developing countries, Africa) |

1. What are the barriers to diagnosing NCDs focusing on depression, hypercholesterolemia, hypertension, obesity and diabetes among PLWH in LMICs in Africa?

2. What are the facilitators of diagnosing NCDs focusing on depression, hypercholesterolemia, hypertension, obesity and diabetes among PLWH in LMICs in Africa?

## Step 2. Identifying relevant studies

**Information sources.** The authors will conduct a comprehensive systematic search within the following databases for a comprehensive search: PubMed, MEDLINE with full text via EBSCO host, Google Scholar, Science Direct and Scopus. The study will use a complete search strategy that employs medical subject headings (MeSH) and keywords. The Boolean terms (AND and OR) will be used to separate keywords, and Medical Subject Headings terms (MeSH) will be conducted in advanced searching of articles. A secondary search of relevant articles from reference lists of included studies using a snowball approach will also be undertaken. All studies with eligible titles will be exported to the EndNote citation manager for deduplication and imported to Rayyan software for systematic review for screening [27]. Two reviewers will independently screen the titles and abstracts based on eligibility criteria, and the third author will be involved in resolving disagreements between the two authors. The PRISMA-Sc guidelines will be used to guide the search in conducting the scoping review. The PRISMA-Sc flow diagram will account for all relevant sources of evidence during screening.

We will perform descriptive data analysis for quantitative studies. The thematic analysis will be carried out using Nvivo version 12 for qualitative studies.

This study will include studies on barriers to and facilitators of NCDs diagnosis, focusing on depression, hypercholesterolemia, hypertension, obesity and diabetes among PLWH in LMICs in Africa.

During the search, we will use the following search terms:

"diagnosis of depression among people living with HIV", "diagnosis of hypercholesterolemia among people living with HIV", "diagnosis of hypertension among people living with HIV obesity", "diagnosis of obesity among people living with HIV", "diagnosis of diabetes people living with HIV", "access to depression care services among people living with HIV", "access to hypertension care services among people living with HIV", "access to diabetes care services among people living with HIV", "access to depression care services among people living with HIV in low—and middle—income countries in Africa/low resource setting" "access to diabetes care services for people living with HIV in low-and-middle-income countries in Africa/low -resource settings", "access to depression intervention services among people living with HIV in low- and middle- income countries in Africa/low-resource setting", "access to obesity intervention services among people living with HIV in low- and middle-income countries in Africa/low-resource setting", "access to hypertension care services for people living with HIV in low-and-middle-income countries in Africa/low-resource settings", "diabetes intervention for people living with HIV in low- and middle-income countries in Africa/low-resource settings"; hypertension intervention for people living with HIV in low- and middle-income countries in Africa/low-resource settings".

A sample search strategy was used (Table 2).

## Step 3. Study selection

The screening process will start with at least two review authors independently assessing the titles and abstracts. Full-text articles will be retrieved for all the titles and abstracts that

**Table 2. The sample search strategy used to identify relevant articles using keywords.**

| S. No | Search strategy | Database used | Number of articles |
|---|---|---|---|
| 1 | (Barriers OR Challenges OR influence OR facilitators OR Opportunities OR Enablers) AND (Diagnosis OR access OR early detection OR Screening OR Treating OR management) AND (Noncommunicable disease OR depression OR hypercholesterolemia OR obesity OR diabetes) AND (People Living with Human Immunodeficiency virus OR AIDS OR PLWH OR PLHIV OR HIV OR People Living with HIV OR Human Living HIV) AND ("Low-and middle-income countries" OR Africa OR "developing countries" OR "low resource settings") | PubMed | 1617 |

potentially meet the eligibility criteria. At least two review authors will again independently assess the full-text articles to decide which studies to include in the review. The third author will be involved in solving disagreements between the two authors.

**Eligibility criteria.** This scoping review will consist of the following eligibility criteria to include articles in this scoping review:

- Studies involving People Living with HIV

- Studies that report evidence on the diagnosis of NCDs focusing on depression, hypercholesterolemia, hypertension, obesity and diabetes among PLWH.

- Studies conducted in low- and middle-income countries in Africa.

- All study designs (interventional, observational, mixed study, qualitative and qualitative designs).

- English language publications

- We decided to include studies published between 2013 and 2023 in this scoping review to get recent evidence on NCD, since the World Health Assembly adopted Global NCD Action Plan in 2013 [28].

This scoping review will exclude.

- Articles published in languages other than English will be excluded because of time, funding challenges, and lack of language resources and language skills in the review team to deal with non-English studies and access to non-English specialized databases.

- Study settings other than low- and middle-income countries other than Africa.

- Studies conducted in high-income countries.

- Studies that focus on the diagnosis of NCD other than depression, hypercholesterolemia, hypertension, obesity and diabetes among PLWH.

**Selection process.** The primary author will conduct a thorough title screening using electronic databases guided by eligibility criteria. All relevant articles will be imported into an EndNote library to remove duplicates of articles. Then articles will be uploaded to Rayyan software and shared with the review team for the next stage of the study screening and selection process. Eligibility criteria will be used to develop a screening tool for abstract and full-text screening phases. Two reviewers will independently conduct abstract and full-text screening phases and group them to include categories. Discrepancies in abstract screening will be addressed through discussion by the review team until a consensus is reached. Assistance from the University of KwaZulu-Natal library services will be sought. The PRISMA-Sc flow diagram [29] will be adopted to report the screening results (Fig 1).

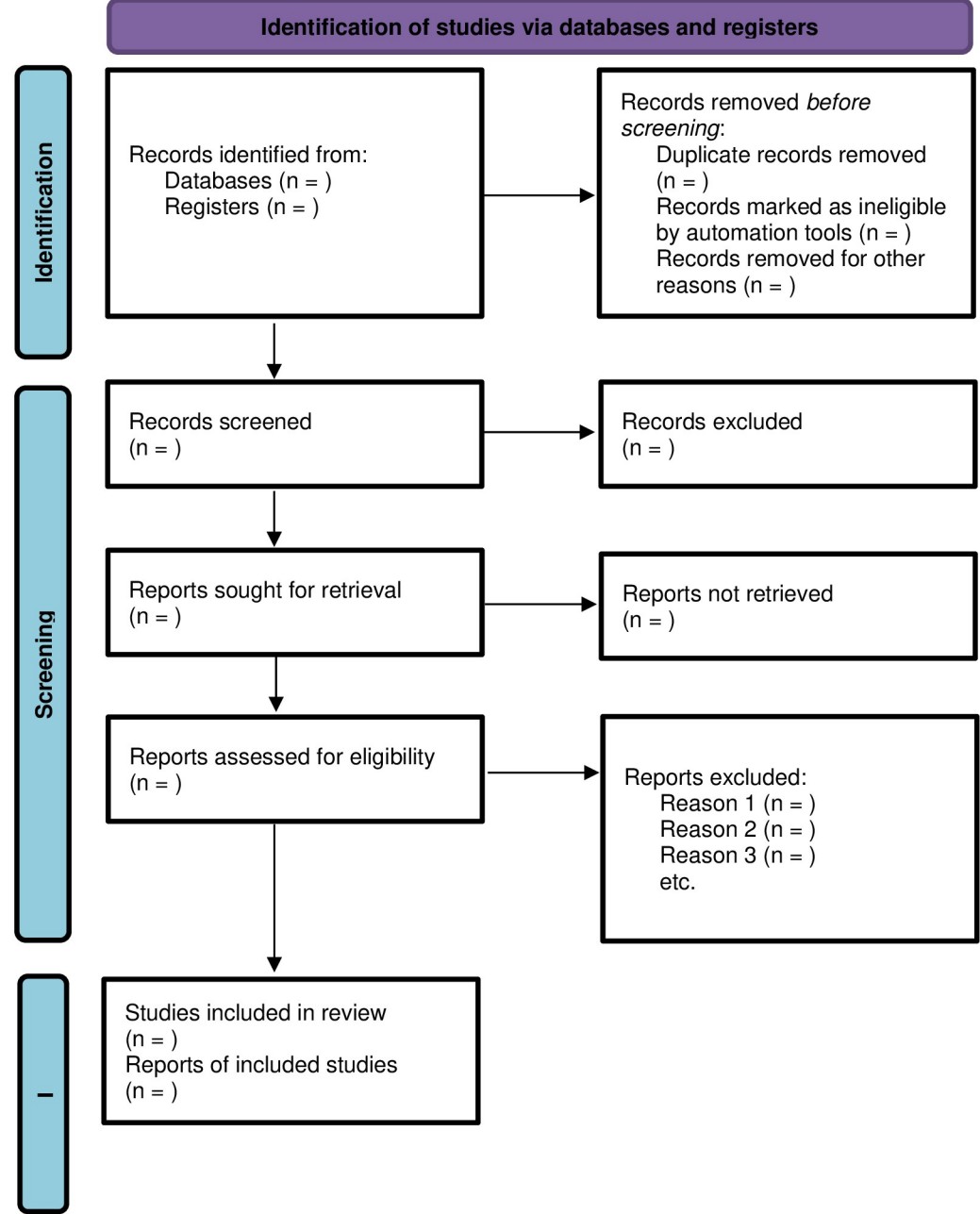

**Fig 1. PRISMA 2020 flow diagram for new systematic reviews, which included searches of databases, registers and other sources.** *From*: Page MJ, McKenzie JE, Bossuyt PM, Boutron I, Hoffmann TC, Mulrow CD, *et al.* The PRISMA 2020 statement: an updated guideline for reporting systematic reviews. BMJ 2021;372:n71. 10.1136/bmj.n71.

## Step 4: Charting the data

A Google form will be developed for data extraction and piloted to ensure accuracy. All relevant data from the included articles will be extracted after thoroughly reading the full texts. Inductive and deductive hybrid approaches will be used to extract data from the included studies for coding and theme development [30]. The data extraction form will consist of the study's

title, author and year of publication, study setting, aims and objectives, country, study design, study participants, type of NCDs, findings relevant to the question (barriers and facilitators of diagnosis of depression, hypercholesterolemia, hypertension, obesity and diabetes), conclusion and recommendations. The data extraction form will be continuously updated to capture all relevant data to answer the review question.

## Step 5: Collating, summarizing, and reporting the results

According to Arksey and O'Malley [24], scoping reviews require thematic frameworks to present narrative accounts of selected literature. The extracted data will be exposed to thematic analysis, and relevant themes and subthemes relating to the study objectives will be developed. In this review, we will use the five steps of thematic analysis recommended by Braun and Clarke [31]. The first step will be data familiarization; the two authors will read through all papers that meet the inclusion criteria in detail. In the second step, initial thematic codes will be inductively generated. The third step will transform the initial codes into emergent themes. In the fourth step, the themes will be reviewed iteratively way to encapsulate all the themes emerging from the entire data set. The fifth will be ensure that the thematic analysis is comprehensive and congruent. NVivo version 12 will be used to assist in thematic data analysis.

## Step 6: Methodological quality appraisal

Although critical appraisal of evidence sources is not mandatory in scoping reviews, the methodological quality of evidence will be assessed using mixed methods appraisal tools [32].

## Discussion

We anticipate that this scoping review will highlight evidence mapping on barriers and facilitators affecting the diagnosis of major NCDs (depression, hypercholesterolemia, hypertension, obesity and diabetes) among PLWH in LMICs. This review will contribute evidence for strengthening and sustaining early diagnosis of hypertension and diabetes among people living with HIV in primary health care facilities. The results of the study will be presented in a tabular form according to the aims of this scoping review. The strengths and limitations of this scoping review method will be detailed. This scoping review may explore different barriers and facilitators of diagnosing NCDs among PLWH and input for strengthened and sustained NCD services for integrated NCD-HIV services in primary health care resource-limited settings for early detection of NCD among PLWH.

## Supporting information

**S1 Data.**
(XLSX)

**S1 Checklist. Preferred Reporting Items for Systematic reviews and Meta-Analyses extension for Scoping Reviews (PRISMA-ScR) checklist.**
(PDF)

## Acknowledgments

The authors intend to acknowledge the University of Kwazulu Natal, South Africa, Health Economics and AIDS research division (HEARD) for giving a PhD scholarship to the primary author. Additionally, the author intends to acknowledge Nontobeko Precious Sikhosana, a professional Librarian University of Kwazulu Natal, for her professional contribution to search

strategies. This scoping review is part of a large-scale study of a PhD project on strengthening ng and sustaining health services for the diagnosis of hypertension and diabetes among PLWH in primary health centres. This scoping review will contribute to a PhD degree award for the primary author.

## Author Contributions

**Conceptualization:** Abebe Sorsa Badacho, Ozayr Harron Mahomed.

**Methodology:** Abebe Sorsa Badacho, Ozayr Harron Mahomed.

**Project administration:** Abebe Sorsa Badacho.

**Supervision:** Ozayr Harron Mahomed.

**Writing – original draft:** Abebe Sorsa Badacho, Ozayr Harron Mahomed.

**Writing – review & editing:** Abebe Sorsa Badacho, Ozayr Harron Mahomed.

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
