## [Decision Letter · Decision Letter 0]

27 Jul 2023

PONE-D-22-28972Mapping evidence on barriers and facilitators of diagnosing hypertension and diabetes among People Living with Human Immune virus (PLWH):  A scoping review protocolPLOS ONE

Dear Dr. Badacho,

Thank you for submitting your manuscript to PLOS ONE. After careful consideration, we feel that it has merit but does not fully meet PLOS ONE’s publication criteria as it currently stands based on my own assessment and the reviewers' reports. Therefore, we invite you to submit a revised version of the manuscript that addresses the points raised during the review process. I recommend you address all the comments raised by the peer-reviewers found below this letter. 

We look forward to receiving your revised manuscript.

Kind regards,

Desmond Kuupiel, PhD

Academic Editor

PLOS ONE

Reviewers' comments:

Reviewer's Responses to Questions

**Comments to the Author**

1. Does the manuscript provide a valid rationale for the proposed study, with clearly identified and justified research questions?

Reviewer #1: Yes

Reviewer #2: Partly

2. Is the protocol technically sound and planned in a manner that will lead to a meaningful outcome and allow testing the stated hypotheses?

Reviewer #1: Partly

Reviewer #2: No

3. Is the methodology feasible and described in sufficient detail to allow the work to be replicable?

Reviewer #1: Yes

Reviewer #2: No

4. Have the authors described where all data underlying the findings will be made available when the study is complete?

Reviewer #1: Yes

Reviewer #2: No

5. Is the manuscript presented in an intelligible fashion and written in standard English?

Reviewer #1: Yes

Reviewer #2: No

6. Review Comments to the Author

You may also provide optional suggestions and comments to authors that they might find helpful in planning their study.

Reviewer #1: Thanks for the opportunity to review this manuscript. The authors detail a scoping review protocol on mapping the evidence of barriers and facilitators of diagnosing hypertension and diabetes among PLWH. The review was well written and studied an important topic. Below are minor suggestions to help improve the manuscript:

- Given your objectives/ research questions within the abstract and methodology, it might be more transparent to add “low resource settings” or “low and middle income countries” within your title.

- On Line 159, it details the “introduction of Art in most developing countries”. Could you please elaborate on this?

- Since you mentioned the use of deductive coding (in addition to inducting coding), it might help to utilize existing frameworks to organize some of the barriers & facilitator themes – e.g. socio-ecological model; or mapping behaviour related facilitators/ barriers to the behaviour change wheel. However, this might be a decision.

Reviewer #2: Reviewer’s Comments

The reviewer would like to thank the author/s for their attempt.

Though this article has the potential to make a significant impact with regards to PLHIV and hypertension and diabetes, the quality of the article is poor. One gets the idea that it was not proof-read and just put together and submitted hoping that the reviewers would edit and fix for the authors.

There are too many spelling and grammatical errors which negate the value in the study.

I believe that at this stage it should be rejected. The authors can then revisit and revise and bring up to publishable standard.

Abstract

The entire abstract needs to be proof-read and revised. Several typo and grammatical errors are evident across the abstract. The acronyms are unnecessarily repeated.

HIV (human immunodeficiency virus) – immune human immune deficiency virus

Low-and middle-income

However, there is limited evidence regarding the barriers and facilitators early diagnosis of hypertension and diabetes among people living with the human immune virus (PLWH) in Low and middle income countries (LMIC).

Non-communicable

Is highly rising – does not read well. Could substitute with is increasing/ is increasing at a rapid rate/in on the increase

Sentence needs to be revised as an denotes singular, yet services is plural.

Studies show an integrated non-communicable disease (NCD), and human immune deficiency virus (HIV) services improved the patient outcome of People living with human immune virus (PLWH) comorbidities with non-communicable diseases (NCDs).

une virus (PLWH) low resource settings. Word missing.

Non-communicable diseases (NCDs) represent a global public health challenge in all population groups, but the prevalence of hypertension and diabetes is highly rising among people living with HIV (PLWH). Studies show an integrated non-communicable disease (NCD), and human immune deficiency virus (HIV) services improved the patient outcome of People living with human immune virus (PLWH) comorbidities with non-communicable diseases (NCDs). It requires a strengthened and sustainable way of diagnosing hypertension and diabetes early among People living with the human immune virus (PLWH). However, there is limited evidence regarding the barriers and facilitators early diagnosis of hypertension and diabetes among people living with the human immune virus (PLWH) in Low and middle income countries (LMIC). This review will systematically map literature and describe the barriers and facilitators of diagnosing of non communicable diseases among people living with the human immune virus (PLWH) low resource settings.

Non-communicable diseases (NCDs) represent a global public health challenge in all

population groups, but the prevalence of hypertension and diabetes is highly rising

among people living with HIV (PLWH).

Methods

This study will include people living with HIV diagnosed with common non-communicable diseases focusing on hypertension and diabetes (word/s missing) resource limited setting.

Discussion

There are inconsistencies in this paper as in the Methods section the focus is on common NCDs and in the discussion the focus is on

This study will include people living with HIV diagnosed with common non-communicable diseases focusing on hypertension and diabetes (word/s missing) resource limited setting.

This scoping review will highlight evidence mapping on barriers and facilitators of

diagnosing major Noncommunicable diseases (hypertension and diabetes) among

people living with the human immune virus (PLWH).

Main document

Globally around 40 million people are living with human immune deficiency virus (HIV), and 48 sub-Saharan Africa is home to more than two-thirds (67%) [1, 2]; the number has increased by 49 more than 10 million since 2010 globally[2].

Line 52 Acquired immune deficiency syndrome (AIDS)-related deaths - HIV (human immunodeficiency virus)

Pg 3 Line 63: Inconsistencies in spelling: Earlier in the abstract Low and middle income countries now low- and middle-income

From the onset the author/s have stated that hypertension and diabetes were the common and major NCDS, however, the stats quoted show that depression (24.4%), hypercholesterolemia (22.2%) are in fact higher than hypertension (21.2%) and diabetes (1.3-18%). This renders the rationale for this scoping review problematic.

Pg 4 A systematic review and meta-analysis of studies conducted in low- and middle-income

64 countries (LMICs) showed pooled estimates for the prevalence of NCDs among PLWH; the prevalence of hypertension, hypercholesterolemia, obesity, depression were 21.2%, 22.2%, 7.8%, 24.4%, respectively, and diabetes ranges from 1.3–18%[12

Pg 4 line 69: use of the signifies a word is missing after positive

among the HIV positive [14, 15];

Could delete the - HIV infected people, HIV positive people, HIV positive patients, HIV-seropositive patients.

Line 80: Spelling - with multi-morbidity are considerable

primary health care (PHC)

Methods

Line 96/7: (1) Identify the research question (2) Searching relevant studies(3)

selecting the studies (4) Charting the data (5) collating, summarizing, and reporting results and (6) consulting with relevant stakeholders to inform or validate findings.

Line 110-112: major NCDs (hypertension and diabetes among PLWH) in the

112 Low and middle income countries .

Context – Needs unpacking – There many LMIC globally. Will this scoping review only focus on LMIC countries in Africa?

Context - Low and middle income countries (developing countries, Africa)

Line 119: What are the barriers of diagnosing hypertension and diabetes among PLWH in (LMIC) Low and middle income countries?

Line 120/1 2. What are the facilitators of diagnosing hypertension and diabetes among PLWH in (LMIC) Low and middle income countries?

Line 133: based on the eligibility criteria

Lines 134-6 The Preferred Reporting Items for Systematic Review and Meta-analysis (PRISMA-Sc) guidelines will be used to guide the search in conducting the scoping review. The PRISMA preferred reporting items for systematic review and meta-analysis flow diagram will account for all relevant sources of evidence during screening.

Line 136/7: We will perform a descriptive data analysis for quantitative studies, and NVIVO

software V.11 for qualitative studies and thematic analysis. will be carried out.

Step 3. Study selection

Words are missing and there are different size fonts in this section.

Line 150: meet the defined eligibility criteria

Studies that report evidence on the diagnosis of NCDs

Studies conducted in low and middle income countries

Line 157: English language publications – Please provide justification for the inclusion of English language publications only.

Line 158-159: Different size fonts. ART should be in capitals.

Studies published between 2003 to June 2022 will be included in the review because the first introduction of Art in most developing countries in 2003[28]

Line 164: Studies carried out in high income countries

Line 166: The Primary author – Why capital letter

Selection process

Line 167: Missing word. an EndNote library

It is unclear if there are 3 reviewers or two. This needs to be unpacked so that any reader will understand how many people are in the research team.

Sometimes researcher/s uses PRISMA and in other instances PRISMA-Sc. I appeal to them to be consistent.

Line 192: Spelling - Figure 1. PRISA 2020

The author/s should double check that they are using the correct PRISA 2020 flow diagram for new systematic reviews

Step 5: Collating, summarizing, and reporting the results

Missing word/s -Lines 204/5: According to Arksey and O’Malley[24] scoping review requires thematic framework to present narrative account of selected literature.

According to Arksey and O’Malley[24] a scoping review requires a thematic framework to present the narrative account of selected literature.

According to Arksey and O’Malley[24] scoping reviews require thematic frameworks to present narrative accounts of selected literature.

Lines 209-213: The author/s state who the first author is (AB) and his role however, who will perform the subsequent roles is unclear. This is linked to my earlier comment on the selection process. There are missing words and spelling and grammatical errors.

the first author (AB) will read through all papers that met the inclusion criteria in detail. In the second step,initial thematic codes will be inductively generated. The third step will transform the initial codes into emergent themes. In the fourth step, the themes will be reviewed in an iterative way to encapsulate all the themes emerging from the entire data set. The fifth will be ensureing that the thematic analysis is comprehensive and congruent. NVivo version12 will be used to assist in thematicdata analysis

Line 215/6 - Step 6: Methodological quality appraisal

Critical appraisal of evidence sources is not mandatory in scoping review, but the

methodological quality will be assessed using mixed methods appraisal tools[32].

Critical appraisal of evidence sources is not mandatory in a scoping review, but the methodological quality will be assessed using mixed methods appraisal tools[32].

Though critical appraisal of evidence sources is not mandatory in scoping reviews, the methodological quality of evidence will be assessed using mixed methods appraisal tools[32].

Discussion

Line 218-221 – are repetitive

We anticipate that this scoping review will highlight evidence mapping on the diagnosis of hypertension and diabetes among PLWH. The barriers and facilitators of diagnosing non communicable disease focusing on hypertension and or diabetes will be identified. This review will contribute evidence for strengthening and sustaining early diagnosis of hypertension and diabetes among people living with HIV in primary care facilities.

…will map evidence on the barriers and facilitators affecting the diagnosing of NCD such as hypertension and diabetes among PLHIV.

Lines 223/4: Line needs to be revised as it does not read well.

The strength and limitations of our scoping review method credibility of the results will be detailed.

References: Reference list and intext references should be checked.

organizatio WH; 2021.

Nations U: 2020 Global AIDS Update Seizing the moment Tackling entrenched

inequalities to end epidemics. In.; 2020.

12. !!! INVALID CITATION !!! [12].

7. PLOS authors have the option to publish the peer review history of their article (what does this mean?). If published, this will include your full peer review and any attached files.

Reviewer #1: No

Reviewer #2: No

<quillbot-extension-portal></quillbot-extension-portal>

---

## [Author Response · Author response to Decision Letter 0]

2 Aug 2023

August 2, 2023

PONE-D-22-28972

Mapping evidence on barriers and facilitators of diagnosing hypertension and diabetes among People Living with Human Immune virus (PLWH): A scoping review protocol

PLOS ONE

Subject: A rebuttal letter that responds to each point raised by the academic editor and reviewer(s). 

Editors comment Journal requirements:

Authors response: Dear editor; Thank you so much for your kind review and time to improve the manuscript. We have revised our manuscript according to PLOS ONE’s style requirements.

Authors response: Dear editor; Thank you so much. No data were generated for this scoping review protocol.

Authors response: Dear editor, Thank you so much for supporting information files for this scoping review protocol at the end.

Reviewers' comments:

Reviewer's Responses to Questions

Point-to-point response to reviewer and editor comment.

Comments to the Author

1. Does the manuscript provide a valid rationale for the proposed study, with clearly identified and justified research questions?

Reviewer #1: Yes

Reviewer #2: Partly

Authors' response: Dear Reviewers, thank you for your kind review and time in evaluating our manuscript. We revised the manuscript that identify and justify the research question. Kindly see the revised manuscript. 

2. Is the protocol technically sound and planned in a manner that will lead to a meaningful outcome and allow testing the stated hypotheses?

Reviewer #1: Partly

Reviewer #2: No

Authors response: Dear Reviewers, thank you for your kind review and time to improve the manuscript. In the revised manuscript, we described methods in detail that answer a research question. 

 3. Is the methodology feasible and described in sufficient detail to allow the work to be replicable?

Reviewer #1: Yes

Reviewer #2: No

Authors response: Dear Reviewers, thank you for your kind review and time to improve the manuscript. We tried to describe the methodology as feasible and in sufficient detail to make the work replicable. Please kindly look at the revised manuscript. 

 4. Have the authors described where all data underlying the findings will be made available when the study is complete?

 Reviewer #1: Yes

Reviewer #2: No

Authors' response: Dear Reviewers, thank you so much. In this scoping review protocol, all supportive information is available with the manuscript, and no data is generated in the current stage.

 5. Is the manuscript presented in an intelligible fashion and written in standard English?

 Reviewer #1: Yes

Reviewer #2: No

Authors' response: Dear Reviewers, thank you so much. We revised and edited the language and corrected grammatical and typographical errors using professional language editors. 

6. Review Comments to the Author

Reviewer #1 Comment: Thanks for the opportunity to review this manuscript. The authors detail a scoping review protocol on mapping the evidence of barriers and facilitators of diagnosing hypertension and diabetes among PLWH. The review was well-written and studied an important topic. Below are minor suggestions to help improve the manuscript:

Authors response: Dear Reviewer, thank you for your kind review and time to improve the manuscript. 

Reviewer #1 Comment: Given your objectives/ research questions within the abstract and methodology, it might be more transparent to add “low resource settings” or “low and middle-income countries” within your title.

Authors response: Dear Reviewer, thank you so much. We revised the title to “Mapping evidence on barriers to and facilitators of diagnosing noncommunicable diseases (NCDs) among people living with human immunodeficiency virus (PLWH) in low- and middle-income countries (LMICs) in Africa: A scoping review protocol.”

Reviewer #1 Comment: On Line 159, it details the “introduction of Art in most developing countries”. Could you please elaborate on this?

Authors response: Dear Reviewer, thank you so much. We have revised the manuscript. We decided to include studies published between 2013 and 2023 in this scoping review to get recent evidence on NCD since the World Health Assembly adopted Global NCD Action Plan in 2013.

Reviewer #1 Comment: Since you mentioned the use of deductive coding (in addition to inducting coding), it might help to utilize existing frameworks to organize some of the barriers & facilitator themes – e.g. socio-ecological model; or mapping behaviour related facilitators/ barriers to the behaviour change wheel. However, this might be a decision.

Authors response: Dear Reviewer, thank you so much for your critical comment. We have not decided to use a framework for deductive coding.

Reviewer #2: Reviewer’s Comments

The reviewer would like to thank the author/s for their attempt.

Though this article has the potential to make a significant impact with regards to PLHIV and hypertension and diabetes, the quality of the article is poor. One gets the idea that it was not proof-read and just put together and submitted hoping that the reviewers would edit and fix for the authors.

There are too many spelling and grammatical errors which negate the value in the study.

I believe that at this stage it should be rejected. The authors can then revisit and revise and bring up to publishable standard.

Authors response: Dear Reviewer, thank you so much for your kind review and time in reviewing the manuscript and constructive feedback to improve the manuscript. We appreciate your critical insights.

Reviewer #2: Reviewer’s Comments Abstract

The entire abstract needs to be proof-read and revised. Several typo and grammatical errors are evident across the abstract. The acronyms are unnecessarily repeated.

Authors response: Dear Reviewer, thank you so much. We revised the abstract and addressed the grammatical and editorial errors according to your suggestions. Please see the track changes on the abstract.

Reviewer #2: Reviewer’s Comments HIV (human immunodeficiency virus) – immune human immune deficiency virus

Low-and middle-income

Authors response: Dear Reviewer, thank you so much. We revised it to HIV (human immunodeficiency virus).

Reviewer #2: Reviewer’s Comments However, there is limited evidence regarding the barriers and facilitators early diagnosis of hypertension and diabetes among people living with the human immune virus (PLWH) in Low and middle income countries (LMIC).

Authors response: Dear Reviewer thank you so much. We revised the statement as “However, there is limited evidence regarding the barriers and facilitators of early diagnosis of NCDs (depression, hypercholesterolemia, hypertension, obesity, and diabetes) among PLWH in low- and middle- income countries (LMICs).

Reviewer #2: Reviewer’s Comments : Non-communicable Is highly rising – does not read well. Could substitute with is increasing/ is increasing at a rapid rate/in on the increase

Authors response: Dear Reviewer, thank you so much. We revised as substituting increasing at a rapid rate.

Reviewer #2: Reviewer’s Comments : Sentence needs to be revised as an denotes singular, yet services is plural.

Studies show an integrated non-communicable disease (NCD), and human immune deficiency virus (HIV) services improved the patient outcome of People living with human immune virus (PLWH) comorbidities with non-communicable diseases (NCDs).

Authors response: Dear Reviewer, thank you so much. We revised it according to your comments. 

Reviewer #2: Reviewer’s Comments : une virus (PLWH) low resource settings. Word missing.

Authors response: Dear Reviewer, thank you so much. We revised the sentences as This review will systematically map the literature and describe the barriers to and facilitators of diagnosing NCDs (depression, hypercholesterolemia, hypertension, obesity, and diabetes) among PLWH in LMICs in Africa.

Reviewer #2: Reviewer’s Comments: Non-communicable diseases (NCDs) represent a global public health challenge in all population groups, but the prevalence of hypertension and diabetes is highly rising among people living with HIV (PLWH). Studies show an integrated non-communicable disease (NCD), and human immune deficiency virus (HIV) services improved the patient outcome of People living with human immune virus (PLWH) comorbidities with non-communicable diseases (NCDs). It requires a strengthened and sustainable way of diagnosing hypertension and diabetes early among People living with the human immune virus (PLWH). However, there is limited evidence regarding the barriers and facilitators early diagnosis of hypertension and diabetes among people living with the human immune virus (PLWH) in Low and middle income countries (LMIC). This review will systematically map literature and describe the barriers and facilitators of diagnosing of non communicable diseases among people living with the human immune virus (PLWH) low resource settings.

Non-communicable diseases (NCDs) represent a global public health challenge in all

population groups, but the prevalence of hypertension and diabetes is highly rising

among people living with HIV (PLWH).

Authors response: Dear Reviewer, thank you so much. We revised as

Noncommunicable diseases (NCDs) represent a global public health challenge in all population groups, but the prevalence of major NCDs such as depression, hypercholesterolemia, hypertension, obesity and diabetes are increasing at a rapid rate among people living with human immunodeficiency virus (PLWH). Studies show that integrated NCDs and human immunodeficiency virus (HIV) services have improved the patient outcome of PLWH comorbidities with NCDs. It requires a strengthened and sustainable way of diagnosing major NCDs early among PLWH. However, there is limited evidence regarding the barriers and facilitators of early diagnosis of NCDs (depression, hypercholesterolemia, hypertension, obesity, and diabetes) among PLWH in low- and middle-income countries (LMIC). This review will systematically map the literature and describe the barriers and facilitators of diagnosing major NCDs (depression, hypercholesterolemia, hypertension, obesity, and diabetes) among PLWH in LMICs in Africa.

Reviewer #2: Reviewer’s Comments: 

Methods

This study will include people living with HIV diagnosed with common non-communicable diseases focusing on hypertension and diabetes (word/s missing) resource limited setting.

Authors response: Dear Reviewer, thank you so much. We revised it as “This scoping review will include studies that included PLWH with those diagnosed with major NCDs (depression, hypercholesterolemia, hypertension, obesity, and diabetes) in LMICs in Africa.”

Reviewer #2: Reviewer’s Comments: 

Discussion

There are inconsistencies in this paper as in the Methods section the focus is on common NCDs and in the discussion the focus is on

This study will include people living with HIV diagnosed with common non-communicable diseases focusing on hypertension and diabetes (word/s missing) resource limited setting.

This scoping review will highlight evidence mapping on barriers and facilitators of

diagnosing major Noncommunicable diseases (hypertension and diabetes) among

people living with the human immune virus (PLWH).

Authors response: Dear Reviewer, thank you so much for your kind review. We have addressed the comments; accordingly, please see the track changes in the manuscript of the abstract section. This scoping review will highlight evidence mapping on barriers and facilitators of diagnosing NCDs (depression, hypercholesterolemia, hypertension, obesity, and diabetes) among PLWH in LMICs in Africa.

Reviewer #2: Reviewer’s Comments:

Main document

Globally around 40 million people are living with human immune deficiency virus (HIV), and 48 sub-Saharan Africa is home to more than two-thirds (67%) [1, 2]; the number has increased by 49 more than 10 million since 2010 globally[2].

Authors response: Dear Reviewer, thank you so much for your kind review. We have addressed the comments; accordingly, Please see the track changes. Globally around 40 million people are living with human immunodeficiency virus (PLWH), globally (1, 2). The number PLWH have increased by more than 10 million since 2010 (2). Sub-Saharan African countries are home to more than two-thirds (67%) of PLWH (1, 2).

Reviewer #2: Reviewer’s Comments:

Line 52 Acquired immune deficiency syndrome (AIDS)-related deaths - HIV (human immunodeficiency virus)

Authors response: Dear Reviewer, thank you so much for your kind review. We have addressed the comments, revised as acquired immunodeficiency syndrome (AIDS)- related deaths - human immunodeficiency virus (HIV).

Reviewer #2: Reviewer’s Comments:

Pg 3 Line 63: Inconsistencies in spelling: Earlier in the abstract Low and middle income countries now low- and middle-income

Authors response: Dear Reviewer, thank you so much for your kind review. We have addressed the comments and used low- and middle-income (LMICs) throughout the manuscript for consistency.

Reviewer #2: Reviewer’s Comments: From the onset the author/s have stated that hypertension and diabetes were the common and major NCDS, however, the stats quoted show that depression (24.4%), hypercholesterolemia (22.2%) are in fact higher than hypertension (21.2%) and diabetes (1.3-18%). This renders the rationale for this scoping review problematic.

Authors response: Dear Reviewer, thank you so much for your constructive insight. We have addressed the comments accordingly. We included major NCDs such as depression, hypercholesterolemia, hypertension, obesity and diabetes in our scoping review. Please see the track changes in the manuscript.

Reviewer #2: Reviewer’s Comments: Pg 4 A systematic review and meta-analysis of studies conducted in low- and middle-income

64 countries (LMICs) showed pooled estimates for the prevalence of NCDs among PLWH; the prevalence of hypertension, hypercholesterolemia, obesity, depression were 21.2%, 22.2%, 7.8%, 24.4%, respectively, and diabetes ranges from 1.3–18%[12

Authors response: Dear Reviewer, thank you for your constructive insight. We have decided to include major NCDs such as depression, hypercholesterolemia, hypertension, obesity and diabetes in our scoping review.

Reviewer #2: Reviewer’s Comments: Pg 4 line 69: use of the signifies a word is missing after positive among the HIV-positive [14, 15];

Authors response: Dear Reviewer, thank you so much. We revised the sentence as “Quality of health and life gains made against HIV/AIDS are now under threat as NCDs are significantly increasing among PLWH in LMICs”

Reviewer #2: Reviewer’s Comments: Could delete the - HIV infected people, HIV positive people, HIV positive patients, HIV-seropositive patients.

Authors response: Dear Reviewer, thank you so much for your constructive insight. We have addressed the comments accordingly, and editorial errors are fixed.

Reviewer #2: Reviewer’s Comments: Line 80: Spelling - with multi-morbidity are considerable

primary health care (PHC)

Authors response: Dear Reviewer, thank you so much for your constructive insight. We have addressed the comments accordingly, and editorial errors have been fixed. Please see the track changes in the manuscript.

Reviewer #2: Reviewer’s Comments: Methods

Line 96/7: (1) Identify the research question (2) Searching relevant studies(3)

selecting the studies (4) Charting the data (5) collating, summarizing, and reporting results and (6) consulting with relevant stakeholders to inform or validate findings.

Authors response: Dear Reviewer, thank you so much for your constructive insight. We have addressed the comments accordingly, and editorial errors have been fixed. Please see the track changes in the manuscript.

Reviewer #2: Reviewer’s Comments: Line 110-112: major NCDs (hypertension and diabetes among PLWH) in the 112 Low and middle income countries .

Authors response: Dear Reviewer, thank you so much for your constructive insight. We have addressed comments accordingly and revised them as “major NCDs focusing on depression, hypercholesterolemia, hypertension, obesity and diabetes in LMICs.”

Reviewer #2: Reviewer’s Comments: Context – Needs unpacking – There many LMIC globally. Will this scoping review only focus on LMIC countries in Africa?

Authors response: Dear Reviewer, thank you so much for your constructive insight. We have addressed the comments accordingly. This scoping review will focus on LMIC countries in Africa.

Reviewer #2: Reviewer’s Comments: Context - Low and middle income countries in Africa (developing countries, Africa)

Line 119: What are the barriers of diagnosing hypertension and diabetes among PLWH in (LMIC) Low and middle income countries?

Authors response: Dear Reviewer, thank you so much for your constructive insight. We have addressed the comments accordingly. We have revised it as What are the barriers to diagnosing depression, hypercholesterolemia, hypertension, obesity, and among PLWH in LMICs in Africa?

Reviewer #2: Reviewer’s Comments: Line 120/1 2. What are the facilitators of diagnosing hypertension and diabetes among PLWH in (LMIC) Low and middle income countries?

Authors response: Dear Reviewer, thank you for your constructive insight. We have addressed the comments accordingly. We have revised it as What are the facilitators to diagnosing depression, hypercholesterolemia, hypertension, obesity, and among PLWH in LMICs in Africa?

Reviewer #2: Reviewer’s Comments: Line 133: based on the eligibility criteria

Authors response: Dear Reviewer, thank you so much for your constructive insight. We revised the eligibility criteria. 

Reviewer #2: Reviewer’s Comments: Lines 134-6 The Preferred Reporting Items for Systematic Review and Meta-analysis (PRISMA-Sc) guidelines will be used to guide the search in conducting the scoping review. The PRISMA preferred reporting items for systematic review and meta-analysis flow diagram will account for all relevant sources of evidence during screening.

Authors response: Dear Reviewer, thank you for your constructive insight. We have addressed the comments accordingly.

Reviewer #2: Reviewer’s Comments: Line 136/7: We will perform a descriptive data analysis for quantitative studies, and NVIVO

software V.11 for qualitative studies and thematic analysis. will be carried out.

Authors response: Dear Reviewer, thank you so much for your constructive insight. We have addressed the comments accordingly. We will perform descriptive data analysis for quantitative studies. The thematic analysis will be carried out using NVivo version 12 for qualitative studies. 

Reviewer #2: Reviewer’s Comments: Step 3. Study selection

Words are missing and there are different size fonts in this section.

Authors response: Dear Reviewer, thank you so much for your constructive insight. We revised.

Reviewer #2: Reviewer’s Comments: Line 150: meet the defined eligibility criteria

Authors response: Dear Reviewer, thank you so much for your constructive insight. We revised.

Reviewer #2: Reviewer’s Comments: Studies that report evidence on the diagnosis of NCDs

Studies conducted in low and middle income countries

Line 157: English language publications – Please provide justification for the inclusion of English language publications only.

Authors response: Dear Reviewer, thank you so much for your constructive insight. We have addressed the comments accordingly. Articles published in languages other than English will be excluded because of cost, time, funding challenges, lack of language resources, and language skills in the review team to deal with non-English studies and access to specialized databases.

Reviewer #2: Reviewer’s Comments: Line 158-159: Different size fonts. ART should be in capitals.

Authors response: Dear Reviewer, thank you so much. We revised and fixed errors. 

Reviewer #2: Reviewer’s Comments: Studies published between 2003 to June 2022 will be included in the review because the first introduction of Art in most developing countries in 2003[28]

Authors response: Dear Reviewer, thank you so much for your constructive insight. We have revised. We decided to include studies published between 2013 and 2023 in this scoping review to get recent evidence on NCD, since the World Health Assembly adopted Global NCD Action Plan in 2013.

Reviewer #2: Reviewer’s Comments: Line 164: Studies carried out in high income countries

Authors response: Dear Reviewer, thank you so much. We have addressed the comments accordingly and revised them as studies conducted in high-income countries will be excluded.

Reviewer #2: Reviewer’s Comments: Line 166: The Primary author – Why capital letter

Authors response: Dear Reviewer thank you so much. We revised as “primary author.”

Reviewer #2: Reviewer’s Comments: Selection process

Line 167: Missing word. an EndNote library

Authors response: Dear Reviewer, thank you so much. We revised accordingly.

Reviewer #2: Reviewer’s Comments: It is unclear if there are 3 reviewers or two. This needs to be unpacked so that any reader will understand how many people are in the research team.

Authors response: Dear Reviewer, thank you so much. We proposed three reviewers review the scoping reveiew. 

Reviewer #2: Reviewer’s Comments: Sometimes researcher/s uses PRISMA and in other instances PRISMA-Sc. I appeal to them to be consistent.

Authors response: Dear Reviewer, thank you so much. We have revised and used the PRISMA-Sc in this manuscript and the PRISMA diagram. 

Reviewer #2: Reviewer’s Comments: Line 192: Spelling - Figure 1. PRISA 2020

The author/s should double check that they are using the correct PRISA 2020 flow diagram for new systematic reviews

Authors response: Dear Reviewer, thank you so much. We have revised as PRISMA diagram. 

Reviewer #2: Reviewer’s Comments: Step 5: Collating, summarizing, and reporting the results

Missing word/s -Lines 204/5: According to Arksey and O’Malley[24] scoping review requires thematic framework to present narrative account of selected literature.

According to Arksey and O’Malley[24] a scoping review requires a thematic framework to present the narrative account of selected literature.

According to Arksey and O’Malley[24] scoping reviews require thematic frameworks to present narrative accounts of selected literature.

Authors response: Dear Reviewer, thank you so much. We have revised according to your suggestion as “According to Arksey and O’Malley [24] scoping reviews require thematic frameworks to present narrative accounts of selected literature.”

Reviewer #2: Reviewer’s Comments: Lines 209-213: The author/s state who the first author is (AB) and his role however, who will perform the subsequent roles is unclear. This is linked to my earlier comment on the selection process. There are missing words and spelling and grammatical errors. the first author (AB) will read through all papers that met the inclusion criteria in detail. In the second step,initial thematic codes will be inductively generated. The third step will transform the initial codes into emergent themes. In the fourth step, the themes will be reviewed in an iterative way to encapsulate all the themes emerging from the entire data set. The fifth will be ensureing that the thematic analysis is comprehensive and congruent. NVivo version12 will be used to assist in thematicdata analysis

Authors response: Dear Reviewer thank you so much; we revised.

Reviewer #2: Reviewer’s Comments: Line 215/6 - Step 6: Methodological quality appraisal

Critical appraisal of evidence sources is not mandatory in scoping review, but the

methodological quality will be assessed using mixed methods appraisal tools[32]. Critical appraisal of evidence sources is not mandatory in a scoping review, but the methodological quality will be assessed using mixed methods appraisal tools[32].

Though critical appraisal of evidence sources is not mandatory in scoping reviews, the methodological quality of evidence will be assessed using mixed methods appraisal tools[32].

Authors response: Dear Reviewer, thank you so much. We have revised according to your suggestion as “Though critical appraisal of evidence sources is not mandatory in scoping reviews, the methodological quality of evidence will be assessed using mixed methods appraisal tools[32].”

Reviewer #2: Reviewer’s Comments: Discussion

Line 218-221 – are repetitive

We anticipate that this scoping review will highlight evidence mapping on the diagnosis of hypertension and diabetes among PLWH. The barriers and facilitators of diagnosing non communicable disease focusing on hypertension and or diabetes will be identified. This review will contribute evidence for strengthening and sustaining early diagnosis of hypertension and diabetes among people living with HIV in primary care facilities.

…will map evidence on the barriers and facilitators affecting the diagnosing of NCD such as hypertension and diabetes among PLHIV.

Authors response: Dear Reviewer, thank you so much. We have revised it according to your suggestion. Please see the track change in the manuscript. as “We anticipate that this scoping review will highlight evidence mapping on barriers to and facilitators affecting the diagnosis of major NCDs (depression, hypercholesterolemia, hypertension, obesity, and diabetes) among PLWH in LMICs.”

Reviewer #2: Reviewer’s Comments: Lines 223/4: Line needs to be revised as it does not read well.

The strength and limitations of our scoping review method credibility of the results will be detailed.

Authors response: Dear Reviewer, thank you so much. We have revised it according to your suggestion. Please see the track change in the manuscript.

Reviewer #2: Reviewer’s Comments: References: Reference list and intext references should be checked. organizatio WH; 2021.

Nations U: 2020 Global AIDS Update Seizing the moment Tackling entrenched

inequalities to end epidemics. In.; 2020.

12. !!! INVALID CITATION !!! [12].

Authors response: Dear Reviewer, thank you so much. We have revised it according to your suggestion and fixed errors in the references. Please see the track change in the manuscript.

---

## [Editor Report · Decision Letter 1]

14 Nov 2023

Mapping evidence on barriers to and facilitators of diagnosing  noncommunicable diseases (NCDs) among people living with human immunodeficiency virus (PLWH) in low- and middle -income countries (LMICs) in Africa: A scoping review protocol

PONE-D-22-28972R1

Dear Dr.Badacho,

We’re pleased to inform you that your manuscript has been judged scientifically suitable for publication and will be formally accepted for publication once it meets all outstanding technical requirements.

Kind regards,

Billy Morara Tsima, MD MSc

Academic Editor

PLOS ONE
---

## [Editor Report · Acceptance letter]

20 Nov 2023

PONE-D-22-28972R1 

Mapping evidence on barriers to and facilitators of diagnosing  noncommunicable diseases (NCDs) among people living with human immunodeficiency virus (PLWH) in low- and middle-income countries (LMICs) in Africa: A scoping review protocol 

Dear Dr. Badacho:

I'm pleased to inform you that your manuscript has been deemed suitable for publication in PLOS ONE. Congratulations! Your manuscript is now with our production department. 

Kind regards, 

on behalf of

Dr. Billy Morara Tsima 

Academic Editor

PLOS ONE